# How Does a Smart City Bridge Diversify Urban Development Trends? A systematic Bibliometric Analysis and Literature Study

Dong Qiu [1], Binglin Lv [1], Calvin M. L. Chan [2,*], Yuesen Huang [3] and Kai Si [1]

[1] School of Management, Fujian University of Technology, Fuzhou 350118, China
[2] School of Business, Singapore University of Social Sciences, Singapore 599494, Singapore
[3] Fujian Provincial Erjian Construction Group Co., Ltd., Fuzhou 350118, China
* Correspondence: calvinchanml@suss.edu.sg; Tel.: +65-6248-4446

**Abstract:** In recent years, the smart city concept has developed rapidly and has gradually become the most popular urban concept. However, the advent of the new century has been accompanied by the emergence of many other emerging city concepts. For these emerging urban concepts, such as a resilient city, low-carbon city, sponge city, and inclusive city, it needs to be clarified how these concepts relate to a smart city. In this paper, the scientometrics software Pajek was used to analyze the publication activities of the city concept and two-mode keywords co-occurrence network with cities. Meanwhile, the study also explores these concepts' global development and correlation. Further, it also analyzes the core problems that each city concept addresses through a literature review. It was observed that although the research content of these four city concepts is different from that of smart cities, they are conceptually and technologically connected with them. The development of smart cities can accelerate the smart development of other city concepts. At the same time, it can acquire and absorb more advanced models from other city concepts to enrich itself. The results suggest that the development of city concepts should be more comprehensive to help cities become more inclusive, safe, resilient, and sustainable, which has important implications for urban policy and practice.

**Keywords:** smart city; resilient city; low-carbon city; sponge city; inclusive city; bibliometric

## 1. Introduction

Cities are one of the most significant social innovations in human history, promoting prosperity and human welfare from both material and non-material aspects [1], which has led to a growing desire for our cities to become more sustainable urban forms and has added many determinants. As a result, many urban concepts have been designed, such as a 'circular city' [2], 'sustainable city' [3], 'green city' [4], 'eco-city' [5], and 'smart city' [6]. Most recently, cities have seen considerable socioeconomic development with the advancement of technology that the 21st century has brought about. However, it also brought about various urban shortfalls, e.g., climate change, traffic congestion, energy shortage, and high-tech crime [7,8]. According to the United Nations, by 2050, the cities will accommodate 6.5 billion people, or about 68% of the world's population, and consume 75% of the world's energy resources [9,10]. Cities are threatened by problems caused by rapid population growth in metropolitan areas [11] which require them to be more inclusive, safe, resilient, and sustainable [12]. Over the past few decades, countries around the world have proposed various initiatives conducive to sustainable urban development. These aim to enhance the urban infrastructure and services, create better environmental, social, and economic foundations, and enhance their attractiveness, competitiveness, and creativity [13]. During this period, more and more urban concepts were introduced to the international policy discourse, especially the concept of a 'smart city' [6].

The concept of smart cities originated at the turn of the century. In 1999, Mahizhnan [14] espoused the concept of smart cities for the first time, signifying technological innovation and globalization in urban areas. However, it was the Smart Earth initiative of IBM in 2008 that piqued the interest in smart cities. The concept of a smart city has since grown to be widely used by governments, enterprises, universities, and research institutes, thus far exceeding the sustainable city concept, especially in the field of urban research [15]. Although the sustainable cities concept was developed earlier and continues, sustainability is an important concept of urban development. Compared to smart cities, its connotation is enormous, and almost all cities can be called sustainable cities, whereas the concept of a smart city is novel and relatively more focused, operable, and technical. The principles of it mainly include intelligent buildings, intelligent water systems, intelligent transport, intelligent safety, intelligent energy, and intelligent healthcare [7,16,17]. A smart city is an advanced form of urban informatization, developed based on the environment of the new generation of information technology and knowledge society innovation 4.0, aiming to fully utilize it in all walks of city life. IBM thinks a smart city develops in the case of limited resources through the use of technology to improve urban services, including civil, commercial, transportation, communication, water, resources, and other urban systems. It further emphasizes the mobile Internet, the Internet of Things, big data, cloud computing, information, and communication technology application of the overall intelligent development, thus realizing the informatization, digitalization, and modernization of urban governance. More smart solutions are adopted through information sharing and transmission to comprehensively address the problems faced previously by urban governance, including urban transportation, energy, communication, water supply, and drainage. With the application of the new generation of information technology, the development of smart cities has become an effective way to solve urban shortfalls.

However, although smart cities are the most referred urban concept and have entered the rapid development stage, they are not the only ones. With the advent of the new century, many other urban concepts emerged, such as resilient and sponge cities. Although these concepts are novel and increasingly researched, few studies have explored the relationship between smart cities and these emerging city concepts in depth. Do these new urban development concepts develop independently from smart city construction concepts? Or is their growth related to that of smart cities? Did their emergence promote the development of smart cities, or did smart cities contribute to the rise of these emerging city concepts? All of these questions need to be answered through in-depth literature analysis, which helps to bridge the gaps in the existing literature in the cross-study of multiple urban concepts. Based on comprehensive bibliometrics and an in-depth literature analysis, this paper explores the origins and definitions of five urban concepts: the smart city, resilient city, low-carbon city, sponge city, and inclusive city. At the same time, the direct or indirect correlation between smart cities and the other four city concepts is intensely analyzed. It is found that although the research content of the four city concepts is different from that of smart cities, they are conceptually or technically related to the smart city. The development of the smart city can accelerate the smart development of other urban concepts. Smart cities also acquire and absorb more advanced models from other urban concepts to enrich themselves.

## 2. Materials and Methods

### 2.1. Research Design

This study refers to the work of De Jong et al. [18], Wang et al. [19], and Schraven et al. [13] in evaluating city concepts using bibliometrics technology and further makes a comprehensive analysis of the selected city concepts. The data collected through an intensive literature review were analyzed using Pajek and COOC software. Pajek is a large-scale, Windows-based network analysis and visualization software developed by A. Mrvar and V. Batagelj in 1996. Unlike regular network analysis software that can handle only small amounts of data, Pajek can handle large networks with millions of nodes, breaking the bottleneck. It can extract

many small networks from large networks, which makes the use of classical algorithms easier for more detailed research and to display networks and analysis results through effective visualization [20]. COOC is software used for rapid bibliometrics and data processing, which is only used to process the original data in this paper [21]. The research concept of this study is represented in Figure 1.

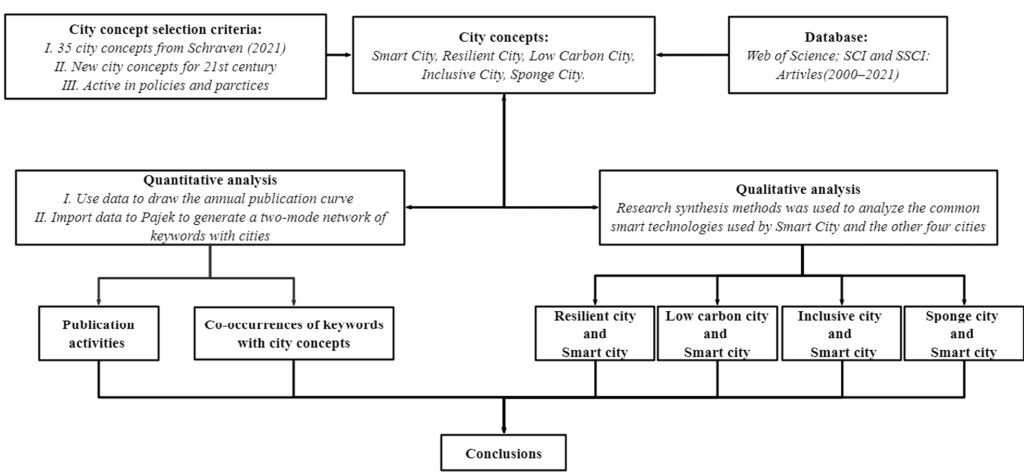

**Figure 1.** Schematic diagram of the research ideas [13].

### 2.2. Data Collection

The research design initiates with a comprehensive data collection accomplished by the following three steps.

Firstly, a search term library should be established to lay the foundation for further analysis [22]. As there are a large number of terms related to urban development, it is crucial to clarify the criteria for selecting city categories in our bibliometric study. Based on the 35 city concepts previously identified [13,18,19,23], this paper adopts the following two criteria to select the city concept to be studied: (1) they must be included in the 35 city concepts as proposed by Schraven et al. [13] and (2) they must be new urban concepts emerging in this century that have been extensively researched and published. That is to say that the whole period of the city concept should be published in more than 50 articles because if the number of articles is less than 50, it is considered to have only made a marginal contribution [13]. (3) They must also be actively used in practice and policymaking worldwide. For example, many countries such as China have put forward initiatives to build sponge cities or low-carbon cities, and the UN SDG11 goal aims to build resilient, inclusive, safe, and sustainable human-inhabited urban communities.

Secondly, to identify the number of publications for the 35 city concepts and minimize the exclusion of the relevant publications in the field, the Web of Science (WoS) Core Collection is chosen as the search engine. As a traditional and comprehensive database for citation analysis, the WoS can support more prolonged periods of citation analysis and high-quality multi-lingual databases compared to other search engines, such as Scopus and Engineering Village [24–26]. Furthermore, the Science Citation Index (SCI) and Social Science Citation Index (SSCI) databases commonly used in bibliometrics were selected to search all 35 city concepts [23]. In addition, only journal articles were analyzed (thus excluding conference proceedings, etc.). After a preliminary literature search, it is found that the concept of emerging cities with increasing influence year by year mainly appeared after the 21st century. In addition, although some urban concepts appeared early, most of them have not received extensive attention from the academic circle, and there are few relevant publications on topics such as an information city, open city, experimental city, etc. Therefore, research articles published between 2000 and 2021 were selected for analysis and retrieved on 29 May 2022. This cohort well represented the distribution of urban concepts in this century, beginning with the first article on the five concepts in 2003 (excluding the 1999

Smart Cities article) until the most recently completed publication at the time of writing. The range of articles encapsulates the essence of the scientific findings and largely reflects the status of city concepts in relevant research fields, those typically used for bibliometric analysis. As some of these resources were irrelevant to the study area, the search results in the WoS were filtered with those pertaining to urban sustainability. Owing to the variability in the city categories (singular, plural, or other variants), this paper adopts a range of search keywords, shown in Table 1, to collect systematic bibliometric data related to the 35 city concepts. Based on these, the four concepts of sponge, low carbon, resilient, and inclusive city were extracted from the thirty-five city concepts. Conforming to the observations of Schraven et al. [13], all four city concepts emerged in the 21st century, and the number of their publications, like smart cities, shows a rapid growth trend. Additionally, the concept of these cities is not only the focus of much theoretical research but is also actively adopted by policymakers and integrated into practice worldwide.

**Table 1.** Details of retrieval settings [26].

| Data Source | Web of Science |
|---|---|
| Search query | TS = ('smart city' or 'smart cities'), TS = ('resilient city' or 'resilient cities'), TS = ('low carbon city' or 'low carbon cities'), TS = ('sponge city' or 'sponge cities'), TS = ('inclusive city' or 'inclusive cities') |
| Document types | DOCUMENT TYPES = (ARTICLE OR REVIEW) |
| Indexes | Indexes = (SCI-EXPANDED, SSCI) |
| Language | Language = English |
| Time span | 1 January 2000 to 31 December 2021 |
| Retrieval time | 29 May 2022 |

A total of 8391 publications on resilient cities, low-carbon cities, sponge cities, inclusive cities, and smart cities were retained for the study, as per the search mentioned through the detailed strategies and standards. The 'Full Record and Cited References' of these resources were exported under the 'plain text' and 'Other Reference Software' file formats to be used for subsequent bibliometric analysis using the Pajek and COOC software tools.

## 3. Results and Discussion

### 3.1. Quantitative Analysis

In this section, the development trend of publications and the map of keywords co-occurrence with cities will be used to shed light on the evolution process of urban concept publications and their discrete characteristics.

#### 3.1.1. Publication Activities of the City Concepts

The number of publications indicates researchers' interest in a particular urban concept at a given time. The development trend of the publications illuminates the research progress of urban concepts and guides the development of future research. To better understand the fundamentals of the five urban concepts in urban research, it is first necessary to understand the details of urban research publishing activities. Figure 2 shows the number of annual publications on sponge, resilient, low-carbon, inclusive, and smart cities obtained from the WoS Core Collection from 1999 to 2021, as well as the quantitative evolution curves of each city concept through the time period. As observed, 'smart cities' are the most published concept so far, followed by 'resilient cities,' 'sponge cities,' and 'low-carbon cities,' with more than 300 publications in each category. 'Inclusive cities' are the least published, with fewer than 100 publications.

It can be deduced from the publication curve in Figure 2 that the concept of the sponge city was first proposed in 2005. From 2006 to 2015, there were no scientific literature publications (except in 2008), indicating that the sponge city was in its infancy and was of little interest to the academic community. It was not until 2016 that the sponge city concept

gradually attracted their attention. Further, since 2017, the curve of the concept of the sponge city has increased significantly, breaking new highs yearly.

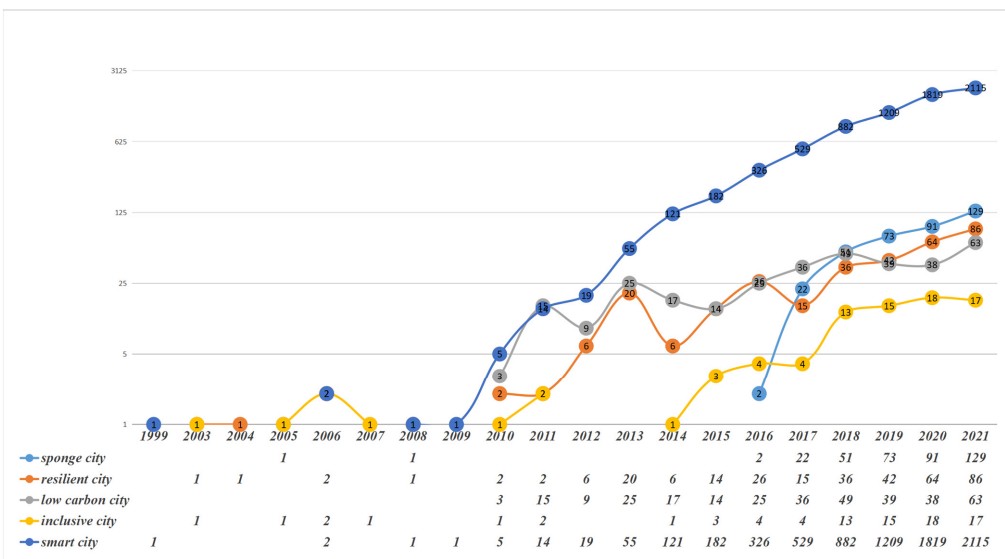

**Figure 2.** Growth trend curve of the number of articles in the 21st century.

The number of research publications on resilient cities increased steadily from 2003 to 2021. The research can be roughly divided into two stages. Stage 1, from 2003 to 2012: the first research on resilient cities was published in 2003, but the number of articles at this stage was only in the single digits and the growth was slow, mainly because the research was in its infancy and received little attention. Stage 2, from 2013 to 2021: relevant research made rapid developments until 2013. However, there was a slight decline in 2014. With the rapid global urbanization process and the surge of various natural and manufactured disasters, cities' vulnerability became more prominent, but the concept caught momentum later in 2021. Therefore, exploring the concept of resilient cities has become a meaningful way to understand sustainable urban development, and the research findings are generally increasing.

Although the inclusive city concept is the same as the resilient city concept, the first publication on it was in 2003, relatively early among the five city concepts. It had a slow start and took some time to gain momentum. Between 2003 and 2014, annual publications on inclusive cities were relatively low, accounting for only 10% of the total number of publications in previous years. It increased marginally from 2015 to 2017 and had a minor peak in 2016. However, it saw exponential growth in the number of publications in 2018 compared to 2017, followed by a record peak in 2020. This indicates that inclusive cities research has received increasing attention in recent years.

The publication of low-carbon cities saw a delayed start, with the first research published in 2010. The research has sustained and grown since then. Although there was a decline in the number of publications in some years, according to the publication trend, this was negated by the significant increase in the overall number of articles.

The earliest paper on smart city research was published in 1999. Until 2009, although a few articles were published occasionally, the growth of the number of articles published during this period was not very significant. There was a negligible increase in the number of articles, and only one research result was published every two years on average during this period. The curve shows exponential growth after 2010. The data shows that more than 70% of the articles were published in the past two years, and more than 90% of the articles were published in the past five years, which indicates that the last five years is the peak period of smart city research, revealing the scientific community's increasing interest

in this topic. As Schraven et al. [13] observed, the research on smart cities is in a stage of rapid growth and has become a vital subject matter in the world's urbanization process.

### 3.1.2. Co-Occurrences of Keywords with City Concepts

To further explore the interconnection between urban concepts, this section illustrates a two-mode city concept and keyword co-occurrence network graph using the Pajek social network analysis tool. The keywords are essential indicators that the authors carefully select to define and represent the direction of their research article, encapsulating the article's basic theory, practical information, and relevance. The network diagram not only shows the characteristics of a single urban concept but also describes the complex conceptual relationship between urban concepts and keywords. For example, identifying the most common keywords in all five city concept articles enables us to gain insight into the content across the five concept fields [13]. Moreover, comparing the keywords across urban concepts can also help reveal their common associations and unique characteristics (the related results are shown in Figure 3).

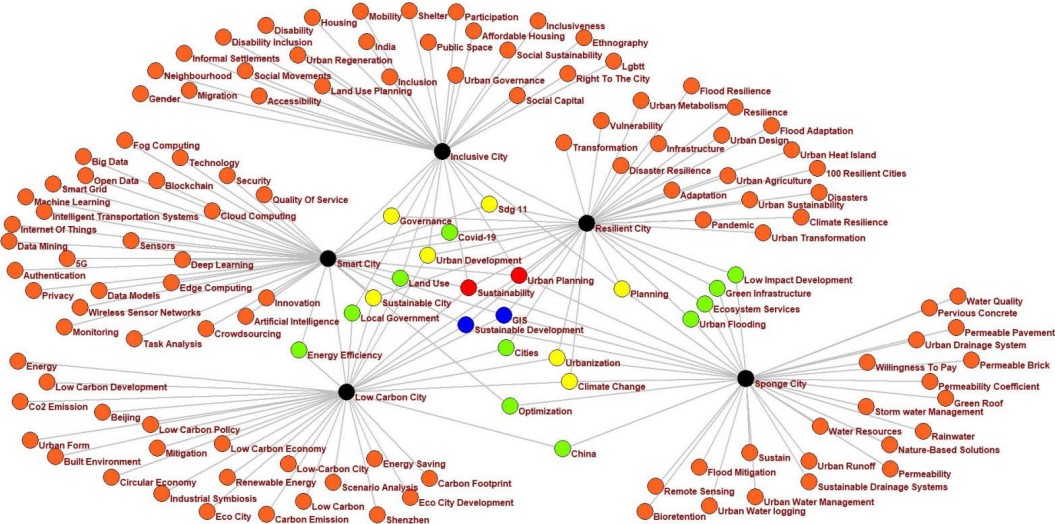

**Figure 3.** The conceptual network structure of keywords (top thirty) associated with the five city concepts.

Figure 3 with the city concepts and keywords two-mode network illustrates the similarities and differences between the concepts. Among them, the red, blue, yellow, and green nodes represent the similarities of five, four, three, and two city concepts' research fields, respectively (i.e., common keywords), while the orange node represents the characteristic keywords unique to each. Each city concept is distinctively marked by a black node. The amount of data is enormous given the 8391 research articles analyzed, producing a total corpus of 20,171 keywords. Therefore, we merge keywords with the same meaning (as shown in Table A1). At the same time, to reduce complexity and ensure the readability of the network graph, only the top 30 high-frequency keywords of each city concept were used in the mapping process. The more lines the keywords radiate, the more they are connected to multiple city concepts and hence, more centrally located (e.g., red, blue, yellow, and green nodes in Figure 3). Conversely, keywords with only a single attachment indicate a unique association with a specific city concept (e.g., the orange node in Figure 3).

#### Common Characteristics

Notably, the network graph shows many similarities, with the red nodes 'Urban planning' and 'Sustainability' positioned in the most central part of the network, together referred to in the top thirty keywords of the five city concepts. This implies that urban

'planning' and 'sustainability' is the central theme of urban development, irrespective of the city concept, and thus plays a crucial role in urban research.

Except for 'Inclusive city', 'Sustainable development' is another common research theme of the other four urban concepts beyond 'Urban planning' and 'Sustainability'. The centrality of the 'GIS' node shows that GIS technology is widely used in the construction and governance process of several cities, such as 'resilient cities', 'smart cities', 'low-carbon cities', and 'sponge cities'. Evidently, the application of smart technology plays a positive role in promoting the rapid development of these several city concepts. In other words, the development of smart cities synonymously promotes the development of a 'resilient city', 'low-carbon city', and 'sponge city.' The nodes 'Governance' and 'SDG11' are both reflected in the three city concepts, 'smart city', 'resilient city', and 'inclusive city', which are closely related to the construction and governance concepts stated by the United Nations Sustainable Development Goals 11. The 'Climate change' node is related to the 'resilient cities', 'low-carbon cities', and 'sponge cities', as observed in the figure.

Since the global COVID-19 outbreak in 2020, countries around the world are actively using information technology for the development and governance of cities, making them more resilient and better able to cope with future natural and man-made disasters. It is not surprising that the node 'COVID-19' appears in the high-frequency keywords of both 'smart city' and 'resilient city'. It is also found that the two city concepts of 'resilient city' and 'sponge city' share relatively many keywords (e.g., 'Low impact development', 'Green infrastructure', 'ecosystem services', and 'Urban flooding'). In particular, the centrality of the node 'Urban flooding' clearly shows that dealing with natural disasters, such as urban floods, is a common problem for 'sponge cities' and 'resilient cities'. However, the difference is that the former mainly focuses on floods as natural disasters with a single research objective, while the latter emphasizes 'disaster resilience', which is not limited to natural disasters and also focuses on developing resilience against man-made disasters. Thus, in broader terms, 'sponge cities' can be viewed as a subset of 'resilient cities'. Additionally, 'resilient city', 'low carbon city', and 'inclusive city' are all connected to 'smart city', which means that there is research relating the smart city concept with these three city concepts. This may explain the frequent creation of terms such as 'smart-low carbon cities', 'smart-resilient cities', 'smart-resilient cities', and 'smart-inclusive cities'.

It is to be noted that the research was based on data in English, which has limited its scope. Therefore, although no connection between the 'smart city' and the 'sponge city' appears in Figure 3, this does not establish that they are not related. For instance, in the Chinese database CNKI, many studies show a specific correlation between 'smart city' and 'sponge city', and terms such as 'smart-sponge city' and 'sponge-smart city' frequently appear.

Individual Characteristics

In addition to the above generic keywords, many specific keywords in the overall network's peripheral regions of each city concept can provide differentiation for a specific city concept.

Most of the keywords of 'smart cities' are related to information technology, which can be roughly divided into three categories. (1) Those that focus on smart transportation application scenarios, including the facilitation of urban transportation and the application of internet and infinite sensor network technology. These include 'intelligent transportation systems', 'wireless sensor networks', 'blockchain', 'internet of things', '5G', and 'monitoring'. (2) Those that focus on the technical dimensions of computing, artificial intelligence, and data analytics. These include 'cloud computing', 'fog computing', 'edge computing', 'data mining', 'big data', 'deep learning', and 'machine learning'. (3) Those with an emphasis on issues such as 'security', 'quality of service', 'innovation', and 'privacy' that are related to the application of technology. From this series of keywords, we can deduce that the rapid development of a 'smart city' seems to be related to the popularization of information technology and its application in daily human life [27,28].

The key attributes of 'resilient cities' can be divided into two categories: (1) those that mainly focus on the resilience construction during disaster prevention and mitigation, such as 'disaster resilience', 'climate resilience', 'pandemic', 'urban heat island', and 'flood resilience' and (2) those that emphasize the concept and analysis of resilient cities, such as 'infrastructure', 'urban design', 'urban transformation', and 'adaptation'.

The characteristics of 'low-carbon cities' can be divided into three categories. (1) Those focused mainly on energy conservation, emission reduction, and other low-carbon city measures, such as 'carbon emission', 'CO2 emission', 'energy saving', and 'renewable energy'. (2) Those focused on 'low carbon economy', 'low carbon police', 'circular economy', 'low carbon development', and other urban low-carbon policies and low-carbon economic system construction. (3) Chinese cities, such as 'Beijing' and 'Shenzhen', which have repeatedly appeared in the keywords. This could be perhaps because China is the world's largest carbon-emitting country. It has set targets of a carbon peak by 2030 and becoming carbon neutral by 2060 at the national level. This could be a reason for the relatively greater number of research cases of low-carbon cities in China.

'Sponge City' keywords can be divided into two categories: (1) those mainly focused on managing water resources in urban areas, such as 'water quality', 'storm water management', 'urban water management', 'urban water logging', and 'flood mitigation' and (2) those that mainly focus on research in the field of water permeability, water storage, drainage, and other related materials and technologies in the city, such as 'remote sensing', 'previous concrete', 'permeable brick', and 'urban runoff.'

As can be seen in the figure, the research focus on 'inclusive city' is mainly divided into two categories: (1) those that focus on prevalent social issues (e.g., 'public space', 'neighborhood', 'housing', and 'land-use planning') and marginalized social groups (e.g., 'lgbtt', 'gender', 'migration', 'ethnography', and 'disability') and (2) those that mainly discuss the 'social capital', 'urban governance', 'participation', 'urban regeneration', and other broad theoretical concepts to deal with the above prevalent issues.

### 3.2. Qualitative Analysis

The above bibliometric research findings show the distinct and common characteristics of the five studied city concepts. To gain better insights into the specific technologies and applications connecting the smart city concepts and the other four cities, a qualitative review of the literature is conducted using the Research Synthesis Method. The Research Synthesis Method conducts a systematic literature analysis based on previous studies to generate new knowledge and interpretation. Table 2 shows the main common technologies and application scenarios of smart cities and the other four major cities.

**Table 2.** Common technologies and application scenarios between the smart city and the other four cities.

| City Concepts | Technologies | Application Scenarios |
|---|---|---|
| Resilient city and Smart city | IoT, Big Data | • The IoT and big data analysis improve the efficiency of daily operation and management of infrastructure, as well as the emergency and response capabilities of cities [29]. |
| | ICT | • ICT technology provides government services and information exchanges and integrates various independent systems to cope with disasters. In particular, it can provide solutions to optimize urban operations during a pandemic [30]. |
| | GIS, | • GIS technology was used to build models to determine the regeneration periods of extreme rainfall and flood, and to collect flood vulnerability information on urban buildings and roads, providing a decision-making basis for resilient urban planning [31,32]. |

**Table 2.** *Cont.*

| City Concepts | Technologies | Application Scenarios |
|---|---|---|
| Low-carbon city and Smart city | Big Data, Cloud Computing | • Big data and cloud computing technologies will be used to build smart energy and carbon emission detection and statistics systems for cities and provide energy consumption and carbon emission data for urban decision-makers to formulate energy conservation and carbon reduction policies [33,34]. |
| | ICT, Big Data, Cloud Computing | • Combine the Internet of Things, big data, and cloud computing to develop green and smart systems to help reduce greenhouse gas emissions in urban traffic [35,36]. |
| | GIS | • Combine GIS technology with statistical analysis to provide location-based information for each home to assess demand side consumption and estimate building carbon emissions across the city [37,38]. |
| Inclusive city and Smart city | Big Data, ICT | • Combining big data with ICT can collect citizens' characteristics, whereabouts, behaviors, and opinions to understand citizens' needs and provide citizens with more adequate and responsive urban planning [39]. |
| | GIS, Remote Sensing | • Remote sensing and GIS technologies are used to produce land maps, land use, and land cover changes to measure urban expansion and vegetation loss rates [40,41]. |
| | ICT, Social Media | • Use ICT to increase civic engagement, where citizens can learn from social networks how to use urban infrastructure while also proposing what other actions cities need to take and where helpful information about those actions is needed to adapt urban infrastructure to the needs of their citizens [42]. |
| Sponge city and Smart city | Remote Sensing, GIS | • GIS technology is used to construct a comprehensive rainwater system, which provides information and technical support information for the whole life cycle of the sponge city construction project. The simulation results of the system under different conditions can provide targeted guidance for the work direction of sponge city construction [43,44]. |

### 3.2.1. Resilient and Smart Cities

The word resilience comes from the Latin word 'resilio', which means 'coming back to the original state'. Resilience has initially been researched in engineering mechanics and was later introduced into system ecology by Holling [45], where it extended from natural ecology to human ecology. Holling [45] described resilience as the ability of the system to sustain and adapt despite the constant interactions within the system. Since the 1990s, resilience has been widely applied in public management and was further developed into a popular urban development concept. Although the concept of a resilient city has been researched for some time now, it lacks a unified focus. The notion of 'resilient cities' is focused on the ability of urban governance systems to adapt and repair themselves in the face of natural disasters and emergencies. Resilient cities, as defined by the OECD [46], must not only promote sustainable development, citizen well-being, and inclusive growth, but must also be able to absorb, recover, and respond to potential future economic, environmental, social, and institutional crises. Meerow et al. [47] proposed that a resilient city refers to the ability of the urban system and all its socio-ecological and socio-technological networks to maintain, quickly recover, adapt, and transform the primary system in response to a crisis.

Zhu et al. [48] evaluated and ranked the resilience of 187 smart cities in China and found a significant positive correlation between urban intelligence and resilience. The findings also suggest that, despite regional differences, smart cities have relatively contributed more to making cities further resilient to disasters [48,49]. The resilience of a

smart city refers to the application of information technology and corresponding services to improve the ability of the urban infrastructure system in the management of external disturbances. Resilience is built based on information technologies, such as the IoT, ICT, GIS, and extensive data analysis, and is used to improve the efficiency of daily operations and management of infrastructure and cities' emergency and response capabilities. Smart city infrastructure systems can provide public emergency shelters during disasters, uninterrupted communication and transport, provide medical care to victims, and restore water or power supply promptly [50].

**IoT**: With the help of the Internet of Things (IoT) and other modern technologies, extensive and diverse data related to the condition and performance of infrastructure systems can be collected. These can be processed, integrated, and analyzed during disasters to help cities recover as quickly as possible. The advanced economy of smart cities, which is the foundation of smart infrastructure and efficient government management and development, can also provide strong support for disaster prediction, preparedness, mitigation, and recovery [29].

**ICT**: The resilience of a city is crucial to its safety and sustainability. Cities with e-government capabilities can use ICT to provide government services, exchange information, and integrate various independent systems in the face of disasters [51]. Through e-governance, citizens can avail public services conveniently, efficiently, and transparently, which in turn increases the resilience of cities. The COVID-19 outbreak is, to some extent, a test of a series of smart city technologies. Sharifi et al. [30] reviewed the literature on smart solutions and technologies during the pandemic and established that smart solutions have enhanced planning, absorption, recovery, and adaptation capabilities that promote urban resilience. Evidently, investment in smart city programs can improve planning and readiness. Additionally, adopting smart solutions and technologies can improve the ability of cities to predict pandemic patterns, promote comprehensive and timely responses, minimize or delay the spread of the virus, provide support for overstretched departments, minimize disruptions to the supply chain, ensure the continuity of essential services, and provide solutions for optimizing urban operations. It dramatically reduces the negative impact on the city structure and improves the resilience of the city on multiple fronts. Smart city technologies can improve the resilience of cities to natural, social, and health disturbances. Florence Metropolitan City has adopted innovative and holistic solutions that harness all communications and large-scale multimedia data in smart cities to manage the resilience of urban transportation systems. In case of disasters, this promotes judicious, data-driven, and action-oriented decisions, thus ensuring the safety of urban residents' lives and property. Even in the absence of disasters, the approach can analyze which areas may be at greater risk, inform decision-making at the planning stage, and implement prevention policies [52].

**GIS**: Vietnam has adopted geoinformatic methods, such as hydrological models and GIS technology, to determine the regeneration period of extreme rainfall and floods, infer the degree of future climate impact on cities, further rationalize the allocation and protection of urban land resources, reduce the potential impact of floods on cities, and provide a friendly living environment for urban residents [31,53–55]. Moreover, Aahlaad et al. [32] proposed a flood resilience planning system that combines geographic database modeling and virtual reality techniques to effectively collect information about the vulnerability of urban buildings and roads to flooding. This information can assist urban policymakers to provide more effective decision-making for the planning and development of resilient cities. Simultaneously, during disasters, it can provide residents with timely and effective disaster prevention information, reduce the impact of floods on urban residents, and safeguard people's lives and property.

It would seem that smart cities have a significant impact on improving urban resilience. They provide smart solutions for urban construction and are critical factors in addressing some significant social challenges such as overpopulation, traffic, pollution, sustainability, safety, health, and the foundation of new companies [56]. Smart cities with the application

of big data, the Internet of Things, and other information technologies enhance urban resistance to natural and man-made disasters and serve as an essential means to boost the optimization and upgrading of cities.

### 3.2.2. Low-Carbon and Smart Cities

Although the concept of low carbon is derived from the word 'low carbon economy' by Kinzig and Kammen [57] in National Trajectories of Carbon Emissions, the influence of the low carbon concept is still in its inception with limited scope. In 2003, the British government first introduced it in its Energy white paper Our Energy Future–Creating a Low Carbon Economy [58]. Thereafter, the concept of a low-carbon economy gained international momentum. The paper proposes increasing economic output by reducing pollution of natural resources and the environment [58]. The low carbon concept then further found social applications, extending from the original low-carbon production to low-carbon life, low-carbon community, and low-carbon city construction. As the primary carrier of greenhouse gas emissions, cities become an essential platform for a low-carbon world. Furthermore, cities are highly centralized, large-scale, and efficient in allocating economic and social resources. Resultingly, low-carbon city construction and its research have since gained impetus. Different scholars since its origin have defined the concept of a low-carbon city from different angles. The World Wide Fund for Nature (WWF) proposed that low-carbon cities are those with low energy consumption and carbon dioxide emissions under the premise of ensuring rapid economic development [59].

Cities efficiently operate, manifesting a high concentration of population, resources, energy, and other factors that further leads to the high concentration of resources, energy consumption, and greenhouse gas emissions. With the advent of the era of 'big data' and the 'Internet of things,' the interactive development of low-carbon cities and smart cities and the construction of lower carbon, greener, smarter, and more convenient cities, are important trends and characteristics of urban development and construction. Low-carbon transformation and smart upgrading of cities are effective ways to deal with climate change, protect the ecological environment, and improve the quality of urban life. On the one hand, the approach to low-carbon cities has stimulated innovative development and application of smart city technologies, and on the other hand, the construction of smart cities paves the way for the development of the low-carbon city. Therefore, it is safe to say that low-carbon and smart cities exist symbiotically. As proposed in the China–US Joint Statement on Climate Change, smart actions to address climate change can stimulate innovation, promote economic growth, achieve sustainable development and energy security, and improve public health and living standards.

The interactive development of the smart city and the low-carbon city further facilitates the interactive development of low-carbon transportation and smart transportation, low-carbon building and smart building, low-carbon industry and smart industry, low-carbon life and smart life, and so on. The new generation of information technology (represented by the Internet of Things, big data, and cloud computing technology) ushers us into the era of novel technological possibilities and economic feasibility.

**Big Data**: Low-carbon cities are long-term investments in urban infrastructure to create sustainable and environment-friendly cities, but data collection for low-carbon cities in the traditional sense is a complex task. Nowadays, low-carbon cities can efficiently collect and process data on urban carbon emissions. This is possible through next-generation information technologies that include adopting big data and cloud computing technology to build a smart energy and carbon emission monitoring and statistics system for cities, communities, buildings, and households, and provide real-time energy consumption and carbon emission data for city governments, community managers, building developers, tenants, and community residents. It provides a quantitative basis for the above bodies to formulate energy saving and carbon reduction policies, adopt energy saving and carbon reduction measures, and realize energy saving and carbon reduction in production and

life. Such new technologies can provide a more reliable, scientific, efficient, and low-carbon energy supply model for low-carbon cities [33,34].

Big data and cloud computing technology can promote the realization of real-time dialogue between urban production capacity and energy consuming equipment, thereby ensuring energy consumption for production and life, assuming energy conservation, environmental protection, green low carbon, safety, and stability. For example, the energy consumption of public buildings and residential buildings can achieve peak staggered complementation during both the day and night, the energy consumption of public buildings and commercial buildings can achieve peak staggered complementarity during holidays, industrial parks and urban core areas can achieve peak staggered complementarity, and there are other aspects of improving the efficiency of urban energy utilization, such as reducing the peak and total energy consumption of urban power grids and achieving energy conservation, environmental protection, and green low carbon.

**IoT**: Smart city construction has brought new opportunities for sustainable and low-carbon development of traditional cities. Urban traffic is one of the most significant sources of pollution and noise; hence, decarbonization in this field is increasingly considered an essential means of climate change mitigation. Smart technologies offer an approach to improving urban traffic efficiency and reducing carbon emissions [36]. The Green Smart system was developed in Spain by combining the IoT, big data, and cloud computing to help reduce greenhouse gas emissions from urban traffic. This scheme has effectively reduced gas emissions (CO 31% on average and CO2 13% on average), travel time (31% on average), and fuel consumption (13% on average) for more than 500 urban scenarios [35].

**GIS**: Similarly, reducing building energy consumption in cities is crucial to overconsumption in low-carbon cities. Italy adopted a framework combining statistical analysis with GIS-based technology to provide location-based information for each residence to assess demand-side consumption within the city limits. Moghadam et al. [37] define various urban transformation energy policies according to local conditions in the smart city environment. Eindhoven used an integrated carbon emission estimation framework based on GIS technology and open data to estimate carbon emissions from buildings and homes. This framework can be used not only to analyze the impact of urban spatial planning on carbon emissions but also for urban optimization and environmental assessment [38].

Understandably, the development of low-carbon cities cannot be separated from the key technical support of smart cities, especially the new generation of information technology to reduce energy consumption and improve efficiency [60]. The use of key technologies such as big data, the Internet of Things, cloud computing, and GIS, can promote the development of green buildings, green communities, green transportation, and other related industries, increase the proportion of green and low-carbon industries, and reduce the utilization rate of traditional energy. With relevant policy mechanisms, the goal of low-carbon development ranging from carbon input to carbon output can be realized.

### 3.2.3. Inclusive and Smart Cities

The inclusive city is a relatively less researched concept, with fewer publications compared to others studied above. Florida [61] was the first to propose the concept of inclusive cities, which, according to the research, is critical to attracting creative talent, supporting high-tech industries, and urban economic growth. The theoretical connotation of inclusiveness is not uniform, and some scholars refer to it as the attitude of individuals or groups toward specific groups or the characteristics of national identity [62]. Lorenz and Schmutzler [63] put forth that an inclusive city means residents' openness to the social, cultural, and ethnic diversity of groups within the region and that where at least two diverse groups coexist and integrate. De Vita and Oppido [64] argue that an inclusive city can be understood as a place where everyone, regardless of gender, age, race, or religious belief, has the right to partake efficiently in the opportunities provided by the city. Similarly, Best and Colman [65] describe inclusive cities that promote participation, inclusion, and

identical citizenship for all citizens. Un-habitat [12] defines inclusive cities as a place where all citizens, regardless of wealth, gender, age, race, and religion, can participate and enjoy the benefits of urban development. It primarily emphasizes the balance and unity of urban development in politics, economy, society, culture, ecological environment, and other arenas that accentuate the inherent consistency of fairness and efficiency in urban development and establishes the homogeneity and equality of development rights of every subject in urban development.

The application of smart technology in inclusive cities enables citizens and city managers to participate in urban governance by applying the new generation of information technologies such as big data, mobile Internet, the Internet of Things, and GIS technology.

**Big Data**: Rebernik et al. [39] suggested that technology can make cities more friendly for those with disabilities and discussed the positive role of existing digital tools in building disability-inclusive cities. Toward this, they introduce mobile phones and mobile applications (e.g., Wheelmap, Lookout, and BlindSquare) to provide accessible navigation for people with disabilities, assist in user-friendly travel and transportation, and provide opportunities for them to participate in collaborative urban governance. Mobile applications, such as Lookout and BlindSquare, based on GNSS, sensors, and visual recognition technology, offer advanced navigation services that help visually impaired people move around by recognizing objects in their surroundings. The study also explores how mobile applications can combine big data with ICT and collect citizens' characteristics, whereabouts, behaviors, and opinions to understand their needs and provide them with more adequate and responsive urban planning.

**GIS**: Urban forests provide myriad benefits to urban life, and the use of urban forest management tools can aid decision-makers in building a more sustainable and inclusive city [66]. In Pakistan, municipalities use drones equipped with high-resolution RGB to count trees in urban areas, calculate the canopy cover index, and estimate the canopy height [41,67]. Moreover, combining remote sensing and GIS technologies can map land use and land cover changes to measure urban sprawl and vegetation loss rates. Pauleit and Duhme [68] combined these technologies to evaluate the spatial pattern and environmental functions of urban forests in Munich, Germany. Narulita et al. [40] investigated the spatial distribution of urban forests in Bandung, Indonesia, and developed a method focusing on the location of the urban forest design.

**ICT**: This new technology enables citizens to be directly involved in the urban decision-making process, mainly characterized by using ICT to increase civic engagement. Citizens can learn from social networks about how to use urban infrastructures, proposing necessary actions for the cities and providing helpful information about those actions needed to adapt the urban infrastructure to the needs of their citizens. Alicante, for example, has adopted new ICT tools (Social Networks Site) to help city decision-making. The analysis of the data on the Social Networks Site can infer urban popular physical exercise areas, assess whether the urban infrastructure is suitable for citizens' sports activities, and provide support to urban policymakers and planners in decision-making, thereby enabling the government and citizens to work together to design a better city and make the city more inclusive [42].

### 3.2.4. Sponge and Smart Cities

Australian researchers first introduced the term 'sponge city' in a population study where they described it as a city that absorbs the surrounding rural population like a sponge [69]. The sponge city concept has been widely recognized in China in recent years. In China, sponge cities were first mentioned in the Low-Carbon City and Regional Development Science and Technology Forum in April 2012 and signified a new urban development plan to manage urban rainwater effectively [70]. In 2013, Chinese President Xi Jinping pointed out at the Central Urbanization Conference that a sponge city is the vision of future urban development and an urban rainwater management system with natural accumulation, natural infiltration, and natural purification. This was the first time that China officially initiated sponge city construction, thus giving rise to a dialogue around

it. Its planning and construction are a practical innovation of the traditional rainwater disposal system and directly contribute to controlling urban drought and flood. The concept focuses on strengthening the supplement and conservation of rainwater to the city's natural water system and groundwater, reasonably increasing rainwater storage and recycling, and setting a moderate-scale drainage pipe network to drain away excess rainwater. Adopting a low-impact development mode to prevent and control waterlogging can effectively reduce the total amount of rainwater entering the drainage pipe network and rainwater runoff per unit of time, thus reducing the pressure and the extensive investment in expanding the drainage pipe network.

Compared with several other city concepts, the smart technology adopted by a sponge city is fairly singular, which only uses GIS and RS technologies to assist the governance of the rain and flood ecosystem in the urban environment.

**GIS**: Zhou et al. [19] integrated the requirements of traditional regulation and sponge city construction and reconstructed a feasible urban rain–flood ecosystem for the new rice region in Yanjin City, China, from four aspects: catchment area and runoff control, road traffic planning, rainwater harvesting and management, and sewage engineering planning. The experimental results showed that the urban rainwater and flood ecosystem integrated with the concept of a sponge city cannot only effectively reduce the flood risk of a smart city, but also improve the construction environment and enhance its vitality. Furthermore, the smart technology based on GIS, remote sensing, and the storm water management model (SWMM) can provide the basis for decision-making for sponge cities' construction projects, such as rainstorm waterlogging prediction and drainage pipe network design. In Xiamen city, China, GIS technology was used to construct the SWMM rain flood model to simulate storm runoff, and the comprehensive benefits of the rain flood model were evaluated by the project. The simulation results of the model include the actual waterlogged areas and are highly consistent with the key investigations of these areas. The simulation results of the model under different conditions can provide targeted guidance for the Yuhong reconstruction project [44]. Chongqing adopted a Uwater-integrated rainwater system based on the GIS platform, which can provide information and technical support for the entire life cycle of sponge city construction projects. When combined with the feedback from the field monitoring system, an adaptive closed-loop system can be formed. Simultaneously, the platform can use GIS spatial data management tools to extract discrete spatial information, such as land use and vegetation cover, and obtain hydrological parameters required for the SWMM simulation, which can be used to evaluate the drainage capacity and corresponding inundation limits of the stormwater system [43].

## 4. Conclusions

In this article, we critically reviewed five urban concepts, including smart cities, resilient cities, low-carbon cities, sponge cities, and inclusive cities, using both bibliometric and in-depth literature analyses.

The study found that the number of academic publications on the five urban concepts has increased considerably over the past 20 years. Although resilient cities, sponge cities, low-carbon cities, and inclusive cities are all emerging concepts in the 21st century, and their publications are very different from those of smart cities, they are still closely related to and integrated with smart cities in terms of theory and technology. The development of smart cities and the popularization of smart technology can empower the smart development of other urban concepts and promote the technological progress of other cities. Smart cities can absorb and learn from the advanced concepts of the other four cities, and thus continuously expand the theoretical connotations of smart cities.

Conclusively, it should be noted that smart cities were initially developed based on smart technology, and their construction concepts are only technology-oriented, but they have since shown strong vitality and inclusiveness. However, there are still many problems in the process of smart city construction. They need to integrate other cities' advanced concepts to make them lower carbon, more inclusive, and more resilient. By

integrating other city concepts, such as the sustainable city, it constantly absorbs advanced experiences of other urban concepts to enrich and evolve itself. At the same time, smart cities reciprocate other urban concepts through their increasingly mature smart technology and further promote the development of other urban concepts. It should be emphasized that both bibliometric analysis and in-depth literature analysis showed that GIS is the most popular and widely used technology in various urban constructions.

The research results effectively offset the shortcomings of existing literature in cross-researching multiple city concepts. The finding also shows that although city literature presents various development paths, their development is not independent and often has a solid connection in technology or concept. This also reminds urban policymakers to be more comprehensive in choosing the concept of cities, making cities more inclusive, secure, resilient, and sustainable. This discovery not only has implications for the theoretical development of the smart city concept but also has practical implications in guiding the trajectory of the development of cities. However, this study also has some limitations. Firstly, the data only analyzed literature in English, which may have an Anglo–American bias. Secondly, the database only selected the WoS, excluding other databases, such as the CNKI Chinese Database. This might have resulted in the loss of some different perspectives and insights. Therefore, future studies may need to include more languages and databases.

**Author Contributions:** D.Q. contributed to the conception of the study and wrote the manuscript; B.L. performed the literature analyses and revised the manuscript; C.M.L.C. edited a portion of the manuscript; Y.H. provided urban development practice projects; and K.S. conducted data collection and collation. All authors have read and agreed to the published version of the manuscript.

**Funding:** The Fujian Institute of Engineering Development Strategies 2021 Key Consulting Projects (project No. 2021-DFZ-20-2). The Innovative Methods Project of the Ministry of Science and Technology of the People's Republic of China (project No. 2020IM010200).

**Institutional Review Board Statement:** Not applicable.

**Informed Consent Statement:** Not applicable.

**Data Availability Statement:** Not applicable.

**Conflicts of Interest:** The authors declare no conflict of interest.

## Appendix A

**Table A1.** A complete list of merged and renamed keywords.

| Renamed Keywords | Included Original Keywords | Reasons for Combination |
|---|---|---|
| Resilient city | Resilient cities | Differences in singular and plural forms but similarities in academic meaning |
| Low carbon city | Low carbon cities; low-carbon city | |
| Sponge city | Sponge cities | |
| Inclusive city | Inclusive cities | |
| Smart city | Smart cities | |
| Green infrastructure | Green infrastructures; Green infrastructure(gi) | Similar academic meaning |
| SDGs | sustainable development goals (sdgs); sustainable development goal; sustainable development goals; sdg; sdgs; sdgs 11 & 10.2; sdg11 | |
| low impact development | low impact development (lid); low impact development; lid | |
| urban flooding | urban flood | |
| urban water logging | water logging | |
| urban water management | water management | |
| internet of things | internet of things (iot); internet of thing; internet of things; iot | |

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
