# Peer review of "How Does a Smart City Bridge Diversify Urban Development Trends? A systematic Bibliometric Analysis and Literature Study"

_sustainability, doi:10.3390/su15054455_

Round 1

Reviewer 1 Report

ear Authors,

I am pleased to review the original manuscript draft entitled "How Does Smart City Bridge Diversified Urban Development Trends: A Systematic Literature Analysis" dedicated to the conceptualisation advances of the smart city concept. The topic is far from new and original, however, this paper could be a great extent to the literature, since it summarises and articulates the notions of the smart city construct. Authors have critically reviewed five urban concepts, including smart cities, resilient cities, low-carbon cities, sponge cities, and inclusive cities, through both bibliometric and in-depth literature analysis.

However, there is room for improvement. I would invite authors to follow a few of my points below:

1. The abstract seems disbalances. A long introduction to the problem, but research objectives, methods, results and implications are not clearly stated. The abstract is a beet messy and difficult to navigate. I propose the restructuring and make an accent on aims, methods, results, and implications.

2. (33-34) the introduction is started with a quite deep statement, it needs to be explained right away. Please add 2-3 following sentences, articulating the statement.

3. (45-48) "more and more" should be accompanied by citations.

4. I would recommend the following authors for inclusion in introduction, using the recent works:

Glebova, E. and Lewicki, W., 2022. Smart cities’ digital transformation. Smart Cities and Tourism: Co-Creating Experiences, Challenges and Opportunities548. Goodfellow Publisher, Oxford.

Angelidou, M., Politis, C., Panori, A., Barkratsas, T. and Fellnhofer, K., 2022. Emerging smart city, transport and energy trends in urban settings: Results of a pan-European foresight exercise with 120 experts. Technological Forecasting and Social Change183, p.121915.

5. (72-88) this is a piece of quite complex text, but it seems unstructured and it is not reader-friendly. Could you please find a way how to navigate the reader in this paragraph? Numbers? Subsections?

6. Figure 1 could be improved in quality, please

7. (4:128-129) "most reliable"? This statement requires at least 2 citations, but anyway may raise questions

8. Conclusion should be extended, giving direct and clear answers to all questions raised in the first part of the paper, explaining findings in greater detail.

Author Response

Response to Reviewer 1 Comments

We sincerely thank you for thoroughly examining our manuscript and providing constructive comments to guide our revision. We have tried our best to revise the manuscript according to your kind and construction comments and suggestions. The responses to the comments are given below.  

Point 1: The abstract seems disbalances. A long introduction to the problem, but research objectives, methods, results and implications are not clearly stated. The abstract is a beet messy and difficult to navigate. I propose the restructuring and make an accent on aims, methods, results, and implications.

Response 1: We accepted your suggestion and rewrote the abstract.

Point 2: (33-34) the introduction is started with a quite deep statement, it needs to be explained right away. Please add 2-3 following sentences, articulating the statement.

Response 2: We have accepted your comments and revised the introduction (31-34).

Point 3: (45-48) "more and more" should be accompanied by citations.

Response 3: We have accepted your comments, adjusted the paragraph and added a citations (46-47).

Point 4: I would recommend the following authors for inclusion in introduction, using the recent works: Glebova, E. and Lewicki, W., 2022. Smart cities’ digital transformation. Smart Cities and Tourism: Co-Creating Experiences, Challenges and Opportunities, 548. Goodfellow Publisher, Oxford. Angelidou, M., Politis, C., Panori, A., Barkratsas, T. and Fellnhofer, K., 2022. Emerging smart city, transport and energy trends in urban settings: Results of a pan-European foresight exercise with 120 experts. Technological Forecasting and Social Change, 183, p.121915.

Response 4: Thanks for your advice, we have cited your recommended article in L60.

Point 5: (72-88) this is a piece of quite complex text, but it seems unstructured and it is not reader-friendly. Could you please find a way how to navigate the reader in this paragraph? Numbers? Subsections?

Response 5: We accepted your recommendation and revised this paragraph (75-90).

Point 6: Figure 1 could be improved in quality, please

Response 6: We have already optimized for Figure 1.

Point 7: (4:128-129) "most reliable"? This statement requires at least 2 citations, but anyway may raise questions

Response 7: We have modified the statement to make it more rigorous (130-133).

Point 8: Conclusion should be extended, giving direct and clear answers to all questions raised in the first part of the paper, explaining findings in greater detail.

Response 8: We have revised the conclusion section (638-650).

Finally, we would like to thank you again for taking the time to review our manuscript, and we hope our correction can get your approval.

Reviewer 2 Report

Topic with potential, but few improvements are requested. However, I have the following suggestions:

Please check the Instructions for Authors regarding the number of words for the Abstract.

L 40. Please set the reference (United, 2018; 2020) in the brackets, according to the Instructions for authors.

L 40. What kind of Problems? It must be detailed. Waste management, energy/energy efficiency, sustainable environment development, etc. I suggest checking and referring to Popescu, D.E.; Bungau, C.; Prada, M.; Domuta, C.; Bungau, S.; Tit, D.M. Waste management strategy at a public university in smart city context. J. Environ. Prot. Ecol.17(3), 2016, 1011-1020, and https://doi.org/10.1016/j.scitotenv.2020.137446 and  https://doi.org/10.3390/su142013121

It is advisable in the introduction section to detail the principles underlying a smart city.  I suggest checking and referring to https://doi.org/10.3390/su142013121

L83-88 must be removed. It is not needed to present what next sections are about, as they can be easily checked in the Figure 1 and in the main manuscript. Instead, After L 82, please better highlight the novelty that your paper brings to the field.

L94. Pajek and COOC software needs to be referenced.

L148. Instead of Table 1 I suggest a PRISMA flow chart, which is much more detailed and comprehensive – I suggest checking Page, M.J.; et al. The PRISMA 2020 statement: An updated guide for reporting systematic reviews. Journal of Clinical Epidemiology 2021, 134, 178-189, doi: 10.1016/j.jclinepi.2021.03.001

L148. It is important to explain in the manuscript why the year 2000 was chosen as the starting point for the analysis.

Why was W.o.S chosen as the search engine and not Scopus which has a wider coverage? A proper justification is needed for the choice of database for the study.

Author Response

Response to Reviewer 2 Comments

We sincerely thank you for thoroughly examining our manuscript and providing constructive comments to guide our revision. We have tried our best to revise the manuscript according to your kind and construction comments and suggestions. The responses to the comments are given below.  

Point 1: Please check the Instructions for Authors regarding the number of words for the Abstract.

Response 1: We have reduced the abstract as much as possible, as the journal requires.

Point 2: L 40. Please set the reference (United, 2018; 2020) in the brackets, according to the Instructions for authors.

Response 2: We have corrected that error.

Point 3: L 40. What kind of Problems? It must be detailed. Waste management, energy/energy efficiency, sustainable environment development, etc. I suggest checking and referring to Popescu, D.E.; Bungau, C.; Prada, M.; Domuta, C.; Bungau, S.; Tit, D.M. Waste management strategy at a public university in smart city context. J. Environ. Prot. Ecol., 17(3), 2016, 1011-1020, and https://doi.org/10.1016/j.scitotenv.2020.137446 and  https://doi.org/10.3390/su142013121

Response 3: These questions have been outlined in L37-38 and cite your recommended articles.

Point 4: It is advisable in the introduction section to detail the principles underlying a smart city.  I suggest checking and referring to https://doi.org/10.3390/su142013121

Response 4: We accepted your advice, listed the principles of smart city in L59-60 and cited your recommended articles.

Point 5: L83-88 must be removed. It is not needed to present what next sections are about, as they can be easily checked in the Figure 1 and in the main manuscript. Instead, After L 82, please better highlight the novelty that your paper brings to the field.

Response 5: We accepted your comment to remove the L83-88 and rewrite the L73-91. 

Point 6: L94. Pajek and COOC software needs to be referenced.

Response 6: We have this part in the article, see L99-105.

Point 7: L148. Instead of Table 1 I suggest a PRISMA flow chart, which is much more detailed and comprehensive – I suggest checking Page, M.J.; et al. The PRISMA 2020 statement: An updated guide for reporting systematic reviews. Journal of Clinical Epidemiology 2021, 134, 178-189, doi: 10.1016/j.jclinepi.2021.03.001

Response 7: Considering that the main purpose of Table 1 is to show the retrieval setting details, we believe that PRISMA flow chart does not well reflect the meaning we want to express. Therefore, we only optimized the Table 1, as shown in L156.

Point 8: L148. It is important to explain in the manuscript why the year 2000 was chosen as the starting point for the analysis.

Response 8: We accepted your advice and explain why the year 2000 was chosen as the starting point for our analysis (134-140).

Point 9: Why was W.o.S chosen as the search engine and not Scopus which has a wider coverage? A proper justification is needed for the choice of database for the study.

Response 9: We have explained the reasons for choosing WoS over Scopus (128-131).

Finally, we would like to thank you again for taking the time to review our manuscript, and we hope our correction can get your approval.

Reviewer 3 Report

see my comments

Author Response

Response to Reviewer 3 Comments

We sincerely thank you for thoroughly examining our manuscript and providing constructive comments to guide our revision. We have tried our best to revise the manuscript according to your kind and construction comments and suggestions. The responses to the comments are given below.  

Point 1: In terms of Literature review, I could not see any mention regarding the standard review that author using; either it is PRISMA or ROSES. I would suggest that author clearly stating the standard on literature review clearly or cite if any protocol review that were made prior. And it would help tremendously if author include if any registration number of the review to ensure the review rigor.

Response 1: Given that the paper's structure has referred to the [8,17,18,22], we expect that the paper will require significant revision if PRISMA or ROSES criteria are adopted. Therefore, the result of our discussion is not adopting the PRISMA or ROSES criteria. However, to more accurately express the article's main idea, we changed the title of the article to How Does Smart City Bridge Diversified Urban Development Trends: A systematic Bibliometric Analysis and Literature Study.

Point 2: In terms of citation writing, it seems that the author does not follow the citation writing standards properly. For example, (Du and Joo 2021) should be written as (Du & Joo 2021). Also, for more than 3 authors (Chen, Zhang, and Zhu 2023) should be written as (Chen at al., 2023). Check all for the rest of the citation in this manuscript.

Response 2: Our references have been in the standard citation writing format of Sustainability journals.

Point 3: Citation wise, this articles also don’t comprehensively change the citation format according to MDPI standard. I saw some citation using IEEE format and some using APA format. E.g. in page 19, citation mixed with [17] and some use (Aahlaad et al., 2021). I would like to suggest author to revise this.

Response 3: We have unified the citation format.

Point 4: Most of the statement for example and many more. I just pick up 1 example i.e. …” The study of green design began in the 1990s as a reflective design philosophy for humanity, society, and the environment, as advocated by industrial designers and artists” the authors should write a citation. It’s difficult for me to state all coz no numbering written in the manuscript.

Response 4: We have added references in statements that may have the above problems.

Point 5: It’s covered the review topic as well as used the relevance of the review topic. However, it would be better to explain the gap between existing papers and how this research came to fully fill the gap in knowledge. It should be identified and clearly stated especially in the abstract and also in the introduction.

Response 5: We have accepted your advice and made some modifications to the abstract and introduction.

Point 6: The abstract should indicate the innovation of the work. It should be written as a summary of the study starting from the aims, methodology until finding of this study. Finally, the gaps and limitations, and future research opportunities should be included. All these should clearly be mentioned.

Response 6: We accept your suggestion, but the journal has word requirements for the abstract. Therefore, due to space constraints, our abstracts only focus on the purpose, method, results, and significance. Innovation, gaps and limitations will be discussed in conclusion.

Point 7: The research design, questions, hypotheses and methods should be clearly stated?

Response 7: We have given a brief explanation of these proposals in the last paragraph of the introduction.

Point 8: Figure 1 should be explained a little bit more as what you are explained in Figure 2.

Response 8: There is a difference between Figure 1 and Figure 2. Figure 2 is the research results of this paper, which require detailed interpretation, while Figure 1 is the study design, which corresponds to the article's structure. The first row of Figure 1 corresponds to the Materials and Methods, and the second and third rows correspond to the Results and Discussion.

Point 9: Table 2- Citation should be uniform, in table section some citation using APA format and some using IEEE format. Please check.

Response 9: We have checked Table 2 and unified the citation format.

Finally, we would like to thank you again for taking the time to review our manuscript, and we hope our correction can get your approval.

Reviewer 4 Report

I find the research idea to be very good, interesting and well-executed. Therefore, I have no major comments on how to improve. It is the appropriate level of quality for the paper dealing with literature analysis.

Author Response

We sincerely thank you for thoroughly examining and recognizing our manuscript.

Round 2

Reviewer 1 Report

Dear Authors,

Thank you for the revisions, the manuscript was improved. 

Please completely address point#4 from the first round of revisions. I share the file for your convenience too.

Author Response

Point 4: I would recommend the following authors for inclusion in introduction, using the recent works: Glebova, E. and Lewicki, W., 2022. Smart cities’ digital transformation. Smart Cities and Tourism: Co-Creating Experiences, Challenges and Opportunities, 548. Goodfellow Publisher, Oxford. Angelidou, M., Politis, C., Panori, A., Barkratsas, T. and Fellnhofer, K., 2022. Emerging smart city, transport and energy trends in urban settings: Results of a pan-European foresight exercise with 120 experts. Technological Forecasting and Social Change, 183, p.121915.

Response 4: Thank you for your sharing. We have cited your shared article in L59. 

Reviewer 2 Report

The authors improved their paper.

Author Response

(The authors gave the same response as above.)

Reviewer 3 Report

I have checked the correction submitted by the authors. The author has made corrections as suggested. 

Author Response

(The authors gave the same response as above.)
